# Verproside, the Most Active Ingredient in YPL-001 Isolated from *Pseudolysimachion rotundum* var. *subintegrum*, Decreases Inflammatory Response by Inhibiting PKCδ Activation in Human Lung Epithelial Cells

**DOI:** 10.3390/ijms24087229

**Published:** 2023-04-13

**Authors:** Eun Sol Oh, Hyung Won Ryu, Mun-Ock Kim, Jae-Won Lee, Yu Na Song, Ji-Yoon Park, Doo-Young Kim, Hyunju Ro, Jinhyuk Lee, Tae-Don Kim, Sung-Tae Hong, Su Ui Lee, Sei-Ryang Oh

**Affiliations:** 1Natural Product Research Center, Korea Research Institute of Bioscience and Biotechnology (KRIBB), Cheongju 28116, Republic of Korearyuhw@kribb.re.kr (H.W.R.); mokim@kribb.re.kr (M.-O.K.);; 2Department of Biological Sciences, College of Bioscience and Biotechnology, Chungnam National University, Daejeon 34134, Republic of Korea; 3Department of Anatomy & Cell Biology, Department of Medical Science, College of Medicine, Chungnam National University, Daejeon 35015, Republic of Korea; mogwai@cnu.ac.kr; 4Disease Target Structure Research Center, KRIBB, Daejeon 34141, Republic of Korea; 5Department of Bioinformatics, KRIBB School of Bioscience, University of Science and Technology (UST), Daejeon 34113, Republic of Korea; 6Immunotherapy Research Center, KRIBB, Daejeon 34141, Republic of Korea; tdkim@kribb.re.kr

**Keywords:** chronic obstructive pulmonary disease, *P. rotundum* var. *subintegrum*, iridoid, NF-κB, MUC5AC, EGR-1

## Abstract

Chronic obstructive pulmonary disease (COPD) is a chronic inflammatory lung disease which causes breathing problems. YPL-001, consisting of six iridoids, has potent inhibitory efficacy against COPD. Although YPL-001 has completed clinical trial phase 2a as a natural drug for COPD treatment, the most effective iridoid in YPL-001 and its mechanism for reducing airway inflammation remain unclear. To find an iridoid most effectively reducing airway inflammation, we examined the inhibitory effects of the six iridoids in YPL-001 on TNF or PMA-stimulated inflammation (IL-6, IL-8, or MUC5AC) in NCI-H292 cells. Here, we show that verproside among the six iridoids most strongly suppresses inflammation. Both TNF/NF-κB-induced MUC5AC expression and PMA/PKCδ/EGR-1-induced IL-6/-8 expression are successfully reduced by verproside. Verproside also shows anti-inflammatory effects on a broad range of airway stimulants in NCI-H292 cells. The inhibitory effect of verproside on the phosphorylation of PKC enzymes is specific to PKCδ. Finally, in vivo assay using the COPD-mouse model shows that verproside effectively reduces lung inflammation by suppressing PKCδ activation and mucus overproduction. Altogether, we propose YPL-001 and verproside as candidate drugs for treating inflammatory lung diseases that act by inhibiting PKCδ activation and its downstream pathways.

## 1. Introduction

Chronic obstructive pulmonary disease (COPD) is characterized by chronic bronchitis, mucus hypersecretion, and emphysema. It is predicted to be the third-leading cause of death worldwide by 2030 [1]. Treatment with bronchodilators and inhaled corticosteroids (ICS) has shown good outcomes in managing COPD; however, ICS have adverse effects, such as osteoporosis. Therefore, there is a growing interest in medicinal plants containing anti-inflammatory compounds that can be used safely and effectively for treating COPD [2].

Traditional medicine in Korea and China has used plants belonging to the genus *Pseudolysimachion* as a phytomedicine for treating pulmonary diseases. Previously, we isolated iridoid glycosides with anti-inflammatory efficacy from a species of the same genus [3]; however, the amount of iridoids in the plants was very limited. From additional efforts to find alternatives in the same genus, we found that *P. rotundum* var. *subintegrum* contains a considerable amount of iridoid glycosides. Among the diverse iridoids extracted from the plant, six high-content iridoids were selected (piscroside C, verproside, isovanillyl catalpol, picroside II, 6-*O*-veratroyl catalpol, and catalposide) and used to constitute a drug candidate mixture named YPL-001 [4]. Currently, Yungjin Pharm. Co., Ltd. has developed a COPD treatment using YPL-001 and completed Clinical Trial Phase 2a. However, among the six iridoids in YPL-001, the essential iridoid and its mechanism for reducing airway inflammation are still unclear. Therefore, in order to develop YPL-001 as a safer and more effective drug against COPD, pharmacological and comparative studies on each iridoid component are required.

TNF upregulates the expression of COPD-related inflammatory genes such as interleukin (IL)-6, IL-8, and MUC5AC [5]. TNF overexpression in mouse airway tissues produces the pathological features of COPD [6]. NF-κB is an essential transcription factor in the TNF pathway to regulate inflammatory gene expression [7]. NF-κB activity is markedly increased in bronchial samples from COPD patients [8]. Thus, downregulating the TNF/NF-κB pathway could be a therapeutic strategy for alleviating lung inflammatory diseases such as asthma and COPD [7].

The human protein kinase C (PKC) family consists of 15 isozymes, sub-divided into three classes: classical (α, β1, βII, and γ), novel (δ, ε, θ, and η), and atypical (ζ and λ/i). In pulmonary diseases, the activation of PKCs causes airway inflammation, bronchospasm, and mucous overproduction. Among them, PKCδ seems closely related to the signaling pathway of COPD pathogenesis [9,10,11,12]. PKCδ induces transcriptional activation of the EGR-1 gene via extracellular-regulated kinase 1/2 [13]. EGR-1 is a transcription factor implicated in COPD-associated chronic inflammation [14]. Thus, a compound that suppresses the PKCδ/EGR-1 axis could be used as a therapeutic agent for treating asthma and COPD.

Here, we compared the inhibitory efficacy of six iridoids constituting YPL-001 on mucus overexpression, and showed that verproside most effectively reduces COPD-related inflammation by suppressing the PKCδ activation and its downstream TNF/NF-κB and PMA/PKCδ/EGR-1 pathways.

## 2. Results

### 2.1. Inhibitory Effects of YPL-001 on TNF-Induced MUC5AC Secretion

YPL-001 contains six iridoid glycosides: piscroside C (iridoid **1**), verproside (iridoid **2**), isovanillyl catalpol (iridoid **3**), picroside II (iridoid **4**), 6-*O*-veratroyl catalpol (iridoid **5**), and catalposide (iridoid **6**) (Appendix A). The mixing ratio of each iridoid to produce YPL-001 was determined by the solute ratio of the methanol extract (Appendix A).

MUC5AC is a gel-forming glycoprotein and a major constituent of mucus that protects the respiratory tract epithelium. However, MUC5AC hypersecretion causes dyspnea due to excessive sputum production in the airways of COPD patients [15]. NCI-H292 is a human lung goblet-like epithelial cell line that readily produces mucus proteins such as MUC5AC. In addition, various stimuli can boost this mucus production/secretion in NCI-H292 cells [16,17], mimicking inflammatory conditions in human airway cells [18]. Thus, we choose the NCI-H292 line for our in vitro screen.

In response to TNF, this cell line produces and secretes excessive amounts of MUC5AC into the surrounding medium. We utilized this feature of NCI-H292 to test the anti-inflammatory activity of YPL-001. Since YPL-001 had no cytotoxicity in NCI-H292 (Figure 1A), we measured the in vitro MUC5AC inhibitory activity of YPL-001 using ELISA. YPL-001 significantly inhibited TNF-induced MUC5AC secretion (Figure 1B, arrowhead) in a dose-dependent manner (Figure 1B, asterisks), suggesting that YPL-001 suppresses COPD-associated mucus hypersecretion in human airway cells.

### 2.2. Inhibitory Effects of the Six Iridoids in YPL-001 on the TNF/NF-κB/MUC5AC Pathway in NCI-H292

TNF-induced NF-κB directly regulates MUC5AC expression by binding to the promoter region of the MUC5AC gene [5,19]. Mucus overproduction via the TNF/NF-κB cascade is a pathophysiological symptom of COPD. Previously, we reported that iridoids **1** and **2** reduce TNF-induced MUC5AC secretion, inhibiting both TNF receptor signaling complex (TNF-RSC) formation and NF-κB activity in human airway cells [5,19]. However, it was not determined whether TNF-induced MUC5AC secretion is also inhibited by iridoids **3**, **4**, **5**, or **6**. Furthermore, it was unidentified which iridoid is the most potent inhibitor or which signaling component is inhibited by them.

First, to identify the most effective iridoid in YPL-001, we tested whether each iridoid in YPL-001 suppresses NF-κB transcriptional activity. HEK293T cells were transfected to express the luciferase reporter gene fused to a promoter with NF-κB-binding motifs (NF-κB-*Luc*). All six iridoids and YPL-001 significantly reduced the TNF-induced transcriptional activity of NF-κB (Figure 2B), suggesting that iridoids **3**, **4**, **5**, and **6** also inhibited TNF/NF-κB-mediated inflammation, similar to iridoids **1** and **2**. However, the efficacies of iridoids **3** and **5** were relatively weaker than those of iridoids **1**, **2**, **4**, and **6**.

Next, we compared the suppressive effects of each iridoid on TNF/NF-κB-induced MUC5AC overproduction in NCI-H292 by qPCR and ELISA. Consistent with the suppressive effects of iridoids on NF-κB-*Luc* activity, all iridoids significantly reduced MUC5AC production, both transcriptionally and translationally (Figure 2C and D, respectively). Interestingly, among the six iridoids, the inhibitory effects of iridoids **1**, **2**, and **4** were most prominent (Figure 2C,D; arrowheads). Moreover, iridoid **2** was the most effective inhibitor of MUC5AC secretion (black arrowheads in Figure 2D), having the lowest IC_50_ value (7.1 μM) compared to iridoids **1** or **4** (9.9 and 11.5 μM, respectively; Figure 2E–G).

The marked inhibitory effect of iridoid **2** showed some interesting aspects of its structure-activity relationship. The better inhibitory effect might be due to the two free hydroxyl groups in the phenyl ring and aglycone epoxy linker (-*O*-) compared with methylated iridoids **3**, **4**, and **5**. A similar relationship was observed between iridoid **2** and monohydroxyl iridoid **6**: dihydroxy iridoid **2** > monohydroxy iridoid **6** > dimethyl iridoid **5** (Figure 2C,D). In the case of iridoids **1** and **2**, the lower inhibitory effect of iridoid **1** might be due to steric hindrance of the aglycone acetyl linkage at C-3 and C-10 compared to the aglycone oxygen bridge at C-7 and C-8 of iridoid **2**.

These results indicate that all six iridoids have anti-inflammatory effects in human airway cells by suppressing the TNF/NF-κB/MUC5AC pathway and suggest that iridoid **2** may be the most effective ingredient in YPL-001.

### 2.3. The Suppressive Effects of Iridoids 1, 2, and 4 on the TNF-Stimulated Activity of the MUC5AC Promoter

Iridoids **1**, **2**, and **4** in YPL-001 effectively suppressed the production and secretion of MUC5AC protein, controlled by the TNF/NF-κB pathway in human airway cells (Figure 2). To further confirm this, we tested the suppressive effects of the three iridoids on the activity of the MUC5AC promoter using a luciferase assay in the presence of TNF. HEK293T cells transiently expressing the MUC5AC promoter-*Luc* were treated with 10 µM each of the three iridoids or 20 µM YPL-001, and their luciferase activity was evaluated. Consistent with the results in Figure 2C–G, YPL-001 and the three iridoids significantly suppressed TNF-stimulated MUC5AC promoter activity (Figure 3, white arrows), suggesting that YPL-001 and the three iridoids inhibit TNF/NF-κB-dependent inflammatory gene expression.

In addition to TNF, other stimulants such as EGF, acrolein, PMA, and CSE also exacerbate mucus overproduction in the airways of COPD patients [20]. Therefore, we also tested whether YPL-001 and iridoids **1**, **2**, and **4** suppress mucus overproduction by other stimulants by assaying MUC5AC promoter-*Luc* activity. Strongly triggered MUC5AC promoter activity (Figure 3) by EGF, PMA, or CSE was suppressed by additional treatment of YPL-001 or the three iridoids. The suppressive effect of iridoid **2** on EGF was marginal (Figure 3C). MUC5AC promoter activation by acrolein, a highly toxic compound formed when fats are broken down by overheating, was not suppressed by YPL-001 and the three iridoids (Figure 3, black arrows). Altogether, our data suggest that YPL-001 and its three iridoids have broad inhibitory effects on various airway stimulants.

### 2.4. Iridoids 2 and 5 in YPL-001 Strongly Inhibited PMA/PKCδ-Induced Inflammatory Gene Expression

Interestingly, Figure 3 shows that YPL-001 and iridoids **1**, **2**, and **4** effectively dampened both PMA-induced and CSE-induced activation of the MUC5AC promoter. PMA stimulates PKCδ and positively regulates the expression of the EGR-1 transcription factor in intestinal cells [13]. In addition, CSE or cigarette smoke increases EGR-1 expression for pulmonary inflammation, including MUC5AC production [14,21]. In this context, we reasoned that under PKCδ-activated conditions by PMA, YPL-001 and its iridoid components could suppress the PMA/PKCδ/EGR-1 axis, such as PMA-induced PKCδ activity or PKCδ-induced EGR-1 expression, to reduce MUC5AC production. To verify our hypothesis, we investigated whether YPL-001 and its six iridoids inhibit PMA-induced PKC activation and subsequent inflammatory gene expression.

We first ascertained that PMA was not cytotoxic when combined with YPL-001 at concentrations below 80 μM (Figure 4A). Subsequently, we examined YPL-001′s suppressive effects on the PMA-induced secretion of mucin/cytokines and found that secretion of MUC5AC and IL-8 was reduced in a concentration-dependent manner (Figure 4B,C).

In addition, YPL-001 significantly suppressed increased TNF levels in PMA-stimulated human mononuclear cells (MNCs) or primary human bronchial epithelial cells (HBECs) without affecting cell viability (Figure 4D,F).

The six iridoids, at 10 μM in the presence of PMA, also suppressed the secretion of MUC5AC, IL-8, and IL-6 (Figure 5B–D) without cytotoxicity (Figure 5A). Interestingly, iridoids **2** and **5** exhibited the most prominent inhibitory effects (Figure 5B–D, white and black arrowheads). Note that the inhibitory effects of each iridoid on PMA-induced MUC5AC secretion differed from those on TNF-induced MUC5AC secretion, which was strongly suppressed by iridoids **1**, **2**, and **4** (Figure 2, arrowheads). These results suggest that YPL-001 and its six iridoids may inhibit PMA-induced PKC activation, which is required for the early stages of inflammation.

Next, we examined the inhibitory effects of YPL-001 and its iridoid components on PMA-induced PKC phosphorylation. PKCδ activation by phosphorylation elicits immune responses, including MUC5AC expression [22,23]. Therefore, we investigated whether YPL-001 and its iridoids inhibit PMA-induced PKCδ phosphorylation in NCI-H292. Western blotting using an antibody-detecting phosphorylated PKCδ at threonine 505 (Thr505) showed that PMA strongly induced PKCδ phosphorylation (approximately 5.5-fold) compared to the negative control (Figure 5E, lanes 1 and 2). This PKCδ phosphorylation was considerably suppressed by YPL-001, iridoid **2**, or iridoid **5** (Figure 5E, arrows and arrowheads in lanes 3, 5, and 8). Notably, the efficacy of iridoid **2** was comparable to that of YPL-001 (Figure 5E; lanes 3 and 5). In contrast, iridoids **1** and **6** showed moderate inhibition of PKCδ phosphorylation (Figure 5E, lanes 4 and 9), whereas iridoids **3** and **4** showed no inhibition (Figure 5E, lanes 6 and 7). 

These results suggest that YPL-001 and its iridoid components **2** and **5** effectively inhibit PMA-induced PKCδ activation, strongly implying that iridoid **2** in YPL-001 may play an important role in reducing airway inflammation.

### 2.5. Iridoids 2 and 5 in YPL-001 Inhibit PMA/PKCδ-Induced EGR-1 Activation

The PKCδ/EGR-1 axis is essential for controlling inflammatory responses. In this axis, PKCδ upregulates the transcription of the *EGR-1* gene [13,22,24]. Moreover, increased EGR-1 expression is associated with disease severity in COPD patients [25]. Therefore, we examined the inhibitory effects of YPL-001 and its six iridoids on EGR-1 expression. PMA-treated NCI-H292 showed increased EGR-1 expression (approximately 6.6-fold compared with the negative control; Figure 5F, lanes 1 and 2), which was considerably reduced by the addition of YPL-001 and its six iridoids (Figure 5F, lanes 3–9). Notably, among the six iridoids, the inhibitory effects of iridoids **2** and **5** were the most intense (Figure 5F, lanes 4 and 7; white and black arrowheads). This strong inhibition of iridoids **2** and **5** on EGR-1 expression was consistent with our data showing iridoid **2**- and **5**-mediated inhibition of MUC5AC/interleukins production and PKCδ phosphorylation (Figure 5B–E). Overall, our data indicate that YPL-001 and its iridoids **2** and **5** effectively suppressed inflammatory responses by inhibiting the PKCδ/EGR-1/MUC5AC pathway in human airway cells.

### 2.6. Verproside Inhibits Phospho-Activation of PKCδ In Vitro and In Vivo

Among the six iridoids, iridoids **1**, **2**, and **4** effectively suppressed the TNF/NF-κB pathway (Figure 2B–D). Iridoids **2** and **5** inhibited the PMA/PKCδ/EGR-1 pathway (Figure 5E,F). Iridoid **2** was the only component that strongly and consistently inhibited both TNF/NF-κB- and PMA/EGR-1-induced inflammation in NCI-H292 cells. Previously, we reported that verproside inhibits TNF-induced TNF-RSC formation, resulting in a reduction in NF-κB activity; however, we failed to determine an upstream regulator of TNF-RSC, inhibited by verproside [19]. Interestingly, PKCδ is a common upstream regulator involved in TNF-RSC formation and EGR-1 expression [13,22,24]. This led us to investigate whether verproside selectively suppresses PKCδ activation to inhibit airway inflammation (or mucin overexpression).

First, to test our hypothesis, we performed western blot analysis in NCI-H292 using antibodies to detect phosphorylated/activated forms of PKC proteins and found that verproside specifically suppressed PMA-induced PKCδ activation by phosphorylation at Thr505 in a concentration-dependent manner (Figure 6A, arrowhead). In contrast, the inhibitory effect of verproside on the activation of other PKC isozymes, such as PKCθ, α/βII, and μ, as well as total PKC, was subtle, marginal, or concentration-independent (Figure 6A).

Next, we examined whether verproside effectively suppresses PKCδ activation under in vivo conditions using cigarette smoke (CS)/LPS-exposed COPD-mouse model. The increased levels of phospho-PKCδ, phospho-ERK, and EGR-1 expression in the lungs of COPD mice were suppressed by administering 25 mg/kg of verproside (Figure 6B). Consistently, MUC5AC and TNF levels in the lungs of COPD mice were significantly decreased by treatment with verproside (Figure 6C,E). Notably, the inhibitory effects of verproside on PKCδ activation and mucin secretion in COPD mice were more effective than those of the same amount of theophylline, a known drug for treating COPD (Figure 6B,C,E). To test the regulatory effect of verproside on inflammatory cell influx into the lungs, a histological examination was performed with H&E staining. The expose of CS/LPS visibly induces the influx of inflammatory cells into the peribronchial region of the lungs of mice. Whereas this cell influx was remarkably decreased by oral treatment of 12.5 and 25 mg/kg verproside (Figure 6F).

These results support our hypothesis that verproside inhibits airway inflammation by inhibiting PKCδ activation both in vitro and in vivo.

Since verproside effectively suppressed airway inflammation by inhibiting PKCδ activation, we checked whether verproside exerts its inhibitory effect by physical interaction with PKCδ. Interestingly, in silico molecular docking simulation between PKCδ and verproside suggested a direct interaction between them (Appendix A). Thus, we performed an in vitro kinase assay using purified PKCδ proteins to determine whether verproside inhibits PKCδ enzymatic activity through direct binding. The kinase activity of PKCδ (Figure 6D, bins 1 and 2) was inhibited by additional verproside (Figure 6D, bins 3, 4, and 5), but the inhibitory effect of verproside was not as strong as that observed in PMA-stimulated NCI-H292 cells (Figure 6A). The in vitro inhibitory effect of verproside on PKCδ activity was also weaker than that of staurosporine or rottlerin (known PKC or PKCδ inhibitors) [26]. These results suggested that verproside can directly inhibit PKCδ catalytic activity, but PKCδ may not be the main target due to the low inhibitory effect of verproside.

Our results also suggested that verproside could inhibit other upstream components that activate PKCδ by phosphorylation.

Overall, our data suggest that verproside in YPL-001 is the most effective iridoid for reducing inflammation observed in COPD by inhibiting phospho-activation of PKCδ, one of the most upstream regulators of the NF-κB and EGR-1 pathways (Figure 7).

## 3. Discussion

In COPD, inflammatory responses releasing cytokines and mucus worsen patients’ symptoms [27]. Thus, agents reducing inflammatory responses can be used as drugs to treat COPD. In this sense, ICS are a good treatment for relieving COPD symptoms. However, long-term or high-dose ICS administration frequently results in side effects or reduces its efficacy in patients with COPD [28]. To overcome this obstacle, we searched for safer drug candidates in traditionally used phytomedicines, and found that a crude extract from *P*. *rotundum* var. *subintegrum*, abundantly containing a group of iridoids (**1**–**6**), has potent anti-inflammatory efficacy against pulmonary inflammation. YPL-001 is a combination of these six iridoids and has passed Clinical Trial Phase 2a for use as a possible therapeutic agent against COPD.

YPL-001 consists of piscroside C, verproside, isovanillyl catalpol, picroside II, 6-*O*-veratroyl catalpol, and catalposide. Among them, mechanistic studies on the anti-inflammatory effects in airway cells have been conducted only for piscroside C, verproside, and picroside II (iridoids **1**, **2**, and **4**) [5,19,29]. The anti-inflammatory effects of other iridoids (isovanillyl catalpol, 6-*O*-veratroyl catalpol, and catalposide) in human epithelial cells were unknown. Moreover, it was unidentified which iridoid has the most potent anti-inflammatory efficacy and which signaling mediators are regulated by the iridoid. In our study, we showed that, among the six iridoids present in YPL-001, verproside (iridoid **2**) is the most efficient agent for reducing inflammatory responses in airway cells, and it inhibits the phosphorylation of PKCδ to reduce inflammation by blocking both TNF/PKCδ/NF-κB and PKCδ/EGR-1 pathways in airway cells.

Several inhibitors for various kinases are under development as a treatment of COPD [30,31]; however, PKC inhibitors have been poorly studied for that purpose, although PKCs are important signaling mediators in chronic airway diseases, including COPD [32]. Recently, several studies using inhibitors for PKCζ/PKCδ (MA130, rottlerin, fisetin, or piscroside C) or an inhibitory peptide against PKCδ reported that PKCs can be useful targets for treating COPD or inflammatory lung diseases [5,23,33,34,35]. Interestingly, most of those agents reduce airway inflammation by targeting PKCδ, and our finding is also in line with this. Thus, we propose that targeting PKCδ among PKCs can be a good strategy for developing a treatment for COPD and that YPL-001 and verproside may be solutions for this.

Phosphorylation of PKCδ at Thr505 is required for its full kinase activity [36,37,38], and it can be produced by the autocatalytic activity of PKCδ itself [37]. In our physiological and cellular tests, verproside effectively suppressed PKCδ-Thr505 phosphorylation. So, we expected that verproside might bind to PKCδ to inhibit this process, thereby inhibiting PKCδ activation/activity. However, our in vitro kinase assay revealed that PKCδ might not be a major target for verproside, although verproside weakly suppresses PKCδ activity by direct interaction. Rather, our result suggested that more upstream regulators can be involved in PKCδ-Thr505 phosphorylation, and they may be a direct target for verproside. Several groups have reported that phosphoinositide-dependent kinase 1 (PDK-1) or Src kinase forms PKCδ-Thr505 phosphorylation [36,37,38,39]. Thus, we can assume that verproside could directly target one of these upstream regulators.

Alternately, the inhibitory effect of verproside on PKCδ might be related to phosphorylation at Tyr311/332 of PKCδ, which is catalyzed by Src and facilitates Thr505 autophosphorylation [37]. Thus, we can guess that verproside binding to PKCδ could interfere with this process to suppress Thr505 phosphorylation. We used purified PKCδ enzyme, activated not by Tyr311/332 but by a lipid activator, to assay the inhibitory effect of verproside in test tubes. Perhaps, due to the limitations of the analytical method we used, we could not observe the inhibitory effect of verproside. Further research is required to find more precise targets of verproside.

In patients with COPD, mucus overproduction and hypersecretion in the airway can be deteriorated by various stimulants, such as TNF, EGF, acrolein, PMA, and CSE [40]. We showed that YPL-001 and its iridoids broadly suppress the harmful effects of these stimuli in airway cells, except for acrolein. Acrolein is a highly reactive and toxic compound formed when fats are burned. So, our data showed that YPL-001 and its iridoids have a wide spectrum of inhibitory effects on various airway stimulants.

Verproside has an apparent effect in inhibiting inflammation in airway cells. Thus, it could be used as a single drug to treat COPD symptoms. However, Figure 2, Figure 3 and Figure 5 imply that mixed application of iridoids, such as YPL-001, may be useful in treating COPD symptoms. As individual iridoids differ in their inhibitory efficacy against each stimulant, the combined use of iridoids may have broad anti-inflammatory effects against various stimuli for airway inflammation. For example, our data (Figure 2B–D and Figure 5B–D) showed that iridoids **1** and **4** preferentially suppressed TNF-induced inflammation, while iridoid **5** effectively reduced PMA-induced inflammation. Moreover, we reported that other minor iridoids, present in small amounts in *P*. *rotundum* var. *subintegrum*, such as longifolioside A and 3-methoxy-catalposide, have anti-inflammatory effects on LPS-induced inflammation in macrophages [41,42]. Therefore, combining multiple iridoids and verproside could be a good strategy for developing a COPD remedy or reliever.

Overall, we suggest that YPL-001 and its single component, verproside, could be considered promising therapeutic agents for treating inflammatory lung diseases, including COPD.

## 4. Materials and Methods

### 4.1. Plant Materials and Preparation

Six iridoids, piscroside C (hereafter iridoid **1**), verproside (iridoid **2**), isovanillyl catalpol (iridoid **3**), picroside II (iridoid **4**), 6-*O*-veratroyl catalpol (iridoid **5**), and catalposide (iridoid **6**) were purified as described previously [19] to the purity of >99.5%. The chemical structures of the iridoids are shown in Appendix A. To express the concentration of YPL-001 in μM, we used an average molecular weight of circa 510, calculated from the molecular weights of the six iridoids (piscroside C: MW 534, verproside: MW 498, isovanillyl catalpol: MW 512, picroside II: MW 512; 6-O-veratroyl catalpol: MW 526, catalposide: MW 482). YPL-001 contains 56.2% of verproside as a major component. Detailed information on the mixing ratio of YPL-001 is available in the patents 10-2014-0122656 and EP3586859A1.

### 4.2. Chemicals and Reagents

Recombinant human EGF and TNF were procured from PeproTech (Cranbury, NJ, USA). PMA, acrolein, lipopolysaccharide (LPS, from *E. coli* serotype 0111:B4), theophylline, and dimethyl sulfoxide (DMSO) were purchased from Sigma-Aldrich (St. Louis, MO, USA). Research cigarette 3R4F and cigarette smoke extract (CSE) were obtained from the Tobacco and Health Research Institute (University of Kentucky, Lexington, KY, USA). Anti-phospho-NF-κB (#CST8242) and phospho-PKC antibody sampler kits were obtained from Cell Signaling Technology (#CST9921). Anti-EGR-1 (#sc-101033), anti-tubulin (#05-829), and secondary antibodies were purchased from Santa Cruz Biotechnology (Dallas, TX, USA), EMD Millipore (Billerica, MA, USA)*,* and GenDEPOT (Katy, TX, USA), respectively.

### 4.3. Cell Preparation and Culture

Human NCI-H292 (CRL-1848) and HEK293T (CRL-3216) cells were obtained from the ATCC. Only cells at early passages (7–20) were used. The mononuclear cells (MNCs) isolated from umbilical cord blood (UCB) were collected from healthy women with full-term pregnancies, with the agreement of the mothers, using human lymphocyte separation solution by density gradient centrifugation [43]. The study was conducted in accordance with the Declaration of Helsinki, and the entire experimental procedure was approved by the Institutional Review Board (IRB) (IRB No. P-01–201610-31–002). NCI-H292, HEK293T. MNCs cells were cultured in a growth medium (GM; RPMI-1640 or DMEM) supplemented with 10% FBS and 100 units/mL penicillin plus 100 µg/mL streptomycin (Hyclone, Logan, UT, USA). Primary human bronchial epithelial cells (HBECs) from normal donors were purchased from Lonza (CC-2540, Morristown, NJ, USA) and maintained as previously described [44]. All cell culture experiments were performed at 37 °C with 5% CO_2_.

### 4.4. Cell Viability Assay

Cells were seeded in a GM-containing 96-well plate at 5 × 10^3^ cells/well. After 24 h, the cells were incubated with YPL-001 or with each of the six iridoids in the presence or absence of PMA (50 nM or 100 nM) or TNF (20 ng/mL) for 24 h. Cell viability was measured using the Cell Counting Kit-8 (Dojindo, Santa Clara, CA, USA) [5].

### 4.5. Enzyme-Linked Immunosorbent Assay (ELISA)

Cells were plated into a 24-well plate at 5 × 10^4^ cells/well and then treated with each compound for 2 h. After adding PMA (100 nM) or TNF (20 ng/mL), they were incubated for 24 h. The presence of cytokines in the supernatant was analyzed using human IL-6 and IL-8 ELISA kits (BD Biosciences, San Jose, CA, USA). Human and mouse MUC5AC protein levels were measured using an anti-MUC5AC antibody (#ab3649, Abcam, Waltham, MA, USA)*,* as previously described [5,45]. MNCs and HBECs were seeded at 5 × 10^5^ cells/well in 12-well plate and 5 × 10^4^ cells/well in 24-well plate, respectively. MNCs and HBECs were treated with YPL-001 for 2 h. Then, PMA (50 nM) was incubated on MNCs and HBECs for 1 h or 6 h, respectively. The secretion level of TNF was measured by ELISA (BD Biosciences, San Jose, CA, USA).

### 4.6. Transfection and Sequential Dual-Luciferase Reporter Assay

HEK293T cells were seeded in a 96-well plate at 2 × 10^4^ cells/well. After 24 h, the cells were transfected with reporter plasmids, such as *NF-κB* luciferase reporter [19] or MUC5AC promoter [5] using Lipofectamine^®^ 2000 reagent (Invitrogen, Carlsbad, CA, USA). Transfection and sequential dual-luciferase activity assays were performed as previously described [5,19]. Luciferase activity (firefly luciferase activity normalized to renilla luciferase activity) was represented as a percentage (%) of the control value. The control value for luciferase activity was obtained from cells treated with DMSO as the vehicle.

### 4.7. *Evaluation of mRNA Expression Level*

Quantitative real-time PCR (qRT-PCR) data were obtained using primers for human MUC5AC and *GAPDH*, performed in triplicate, and analyzed by the 2^−ΔΔCT^ method, as previously described [19].

### 4.8. Western Blot Analysis

NCI-H292 cells (5 × 10^5^ cells/well) were seeded in 6-well plates and incubated for 12 h in GM. GM was subsequently replaced with a serum-reduced GM medium (0.1% FBS). After 16 h, the cells were pre-treated with the corresponding concentrations of the compounds for 2 h and subsequently treated with PMA (1 μM) for 30 min. Instead of 100 nM PMA, 1 μM PMA was used for faster and more robust activation of PKCδ and its downstream signaling molecules. Lung tissue lysates were homogenized in a RIPA lysis buffer (1/10 *w*/*v*) containing a protease inhibitor cocktail (Sigma-Aldrich, St. Louis, MI, USA) [10]. Proteins were prepared and loaded as described previously [10,19]. The protein bands were analyzed using Amersham Imager 680 (GE Healthcare, Arlington Heights, IL, USA) and densitometry analysis software (version 2.0.0). The original uncropped images of the western blot are shown in Appendix A.

### 4.9. Cigarette-Smoke(CS) Exposure and LPS Treatment in a Mouse COPD Model

Six-week-old male C57BL/6 mice (20~25 g, Koatech, San Francisco, CA, USA) were whole-body exposed to fresh air or CS (from eight cigarettes, 3R4F research cigarettes; Tobacco and Health Research Institute, University of Kentucky, Lexington, KY, USA) for 1 h/day for seven days using a smoking machine (SciTech, Seoul, Republic of Korea). Research cigarettes (3R4F) containing 11.0 mg of total particulate matter, 9.4 mg of tar, and 0.76 mg nicotine per cigarette were purchased from the University of Kentucky (Lexington, KY, USA). LPS (5 μg dissolved in 50 μL distilled water) was administered intranasally on day six. Verproside and theophylline (an inhibitor of phosphodiesterase 4) were orally administered to the animals 1 h before CS exposure for seven days. Theophylline, a commercial drug for treating chronic inflammatory diseases, was used as a positive control [46]. Mice were randomly divided into four groups (*n* = 6 per group). NC: normal control mice; COPD: CS and LPS-exposed mice; Theophylline: CS and LPS-exposed mice treated with theophylline (25 mg/kg); Verproside (12.5 or 25 mg/kg): CS and LPS-exposed mice treated with verproside (12.5 or 25 mg/kg). No signs of toxicity, such as dyspnea or vomiting, were observed.

### 4.10. Computational Simulations: Homology Modeling and Molecular Docking

Since the structure of human PKCδ has not been characterized, homology modeling was performed to generate the human PKCδ protein structure for molecular docking simulations [23]. The structures of each of the six iridoid compounds were obtained from PubChem (https://pubchem.ncbi.nlm.nih.gov/ accessed on 7 January 2021) and determined using the Marvin program (https://www.chemaxon.com accessed on 10 January 2021). As previously described, the docking simulation between PKCδ and each compound was performed using the Autodock Vina program (http://vina.scripps.edu accessed on 6 February 2021), as presented in Appendix A [23].

### 4.11. Determination of PKCδ Kinase Activity under the Cell-Free Condition

PKCδ activity was assayed using a *PKC*δ kinase assay kit (V3401 and V9101, Promega, Fitchburg, WI, USA). In brief, diluted recombinant full-length human *PKC*δ (100 ng) was mixed with reaction components (0.2 μg/μL substrates and PKC lipid activator) in a white microplate. The mixture was pre-incubated for 10 min at 30 °C in the presence or absence of the test compounds. The amount of DMSO in the reaction solution was set to 4%. The PKCδ reaction was initiated by adding adenosine triphosphate (ATP, 50 μM) and incubated at 30 °C for 15 min. The reaction was terminated by adding ADP-Glo reagent. The final PKCδ activity was assayed by adding kinase detection reagent and analyzed using a luminometer (a SpectraMax^®^ M4 microplate reader, Molecular Devices, San Jose, CA, USA).

### 4.12. Determination of the Levels of TNF in Bronchoalveolar Lavage Fluid (BALF)

After anesthetizing the mice on day 8, BALF was collected. The levels of TNF were determined according to a previously published protocol [47].

### 4.13. Histological Examination

After BALF collection, the lung tissue was fixed using 10% (*v*/*v*) neutral buffered for malin. Formalin-fixed lung tissue was embedded in paraffin, sectioned at 4 μm thickness, and stained with hematoxylin and eosin (H&E) solution to evaluate the levels of inflammatory cell influx [48].

### 4.14. Statistical Analysis

All data are presented as mean ± standard deviation of three independent experiments unless otherwise described. Statistical significance was calculated using Student’s *t*-test. The differences between the compound-treated groups were determined by two-way ANOVA with Tukey’s multiple comparison test using GraphPad Prism (version 6).

## 5. Conclusions

Here, we investigated the mechanism(s) of action of six iridoids constituting YPL-001 on reducing airway inflammation. We propose that YPL-001 and its iridoids could be possible therapeutic agents for reducing airway inflammation. Our results show that YPL-001 and its iridoids suppress the TNF/NF-κB/MUC5AC and PMA/PKCδ/EGR-1 cascades, which are important signaling pathways involved in inflammatory lung diseases, including COPD and asthma. Of the six iridoids, verproside most effectively inhibits both NF-κB and EGR-1 cascades and suppresses the phospho-activation of PKCδ proteins, a common upstream regulator of both signaling cascades. In vitro kinase assay, verproside can bind to PKCδ. However, its interaction with PKCδ is insufficient for a strong reduction of PKCδ activation, suggesting that another signaling component that regulates PKCδ could be a major target for verproside. Taken together, we propose that verproside or the verproside-based mixture of iridoids could be a potential therapeutic agent for managing COPD symptoms.

## Figures and Tables

**Figure 1 ijms-24-07229-f001:**
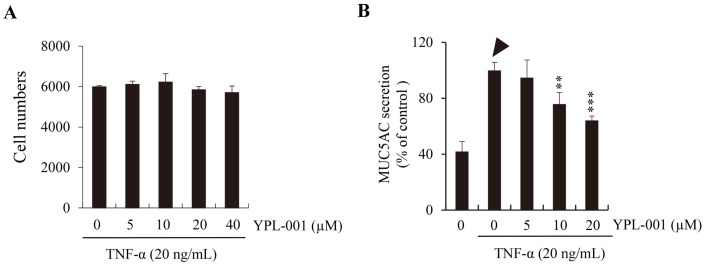
Inhibitory effect of YPL-001 on TNF-induced MUC5AC secretion in NCI-H292 cells. (**A**) No cytotoxicity of YPL-001 in NCI-H292 in the presence of TNF (20 ng/mL). (**B**) Suppressive effect of YPL-001 on the secretion of MUC5AC protein (arrowhead) from TNF-stimulated NCI-H292. ** *p* < 0.01 and *** *p* < 0.001 compared to the TNF alone.

**Figure 2 ijms-24-07229-f002:**
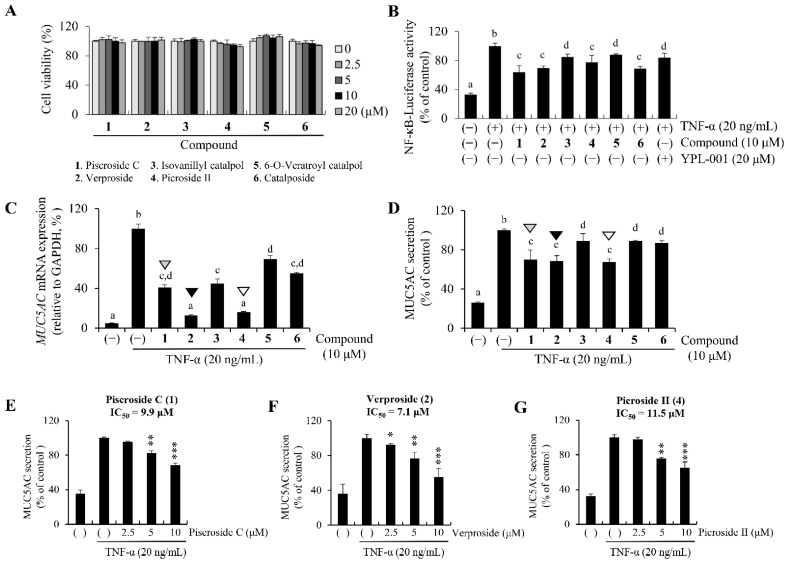
Inhibitory effects of the six iridoids in YPL-001 on TNF/NF-κB-induced MUC5AC expression in NCI-H292. (**A**) No cytotoxicity of six iridoids in NCI-H292. (**B**) Luciferase-based reporter assay tested under the control of NF-κB-responsive elements containing minimal promoter. (**C**,**D**) Suppressive effects of each iridoid (**1**–**6**) on TNF-induced MUC5AC production. MUC5AC mRNA transcription and protein secretion were assayed using qPCR and ELISA, respectively. NCI-H292 were pre-treated with 10 µM of each iridoid for 2 h and then treated with TNF (20 ng/mL) for an additional 12 h to carry out qPCR or for an additional 24 h to carry out ELISA. The three iridoids have strong suppressive effects on TNF-induced MUC5AC expression (piscroside C in grey, verproside in black, and picroside II in white arrowhead). Two-way ANOVA with Tukey’s multiple comparisons test was performed (**B**–**D**). ^a–d^ Different superscript letters on the bar graph mean significant statistical differences (*p* < 0.05). There is no statistical significance between groups sharing the same letter. (**E**–**G**) The half-maximal inhibitory concentration (IC_50_) of iridoids **1** (piscroside C), **2** (verproside), or **4** (picroside II) on MUC5AC secretion. * *p* < 0.05, ** *p* < 0.01, and *** *p* < 0.001, as compared to the TNF alone.

**Figure 3 ijms-24-07229-f003:**
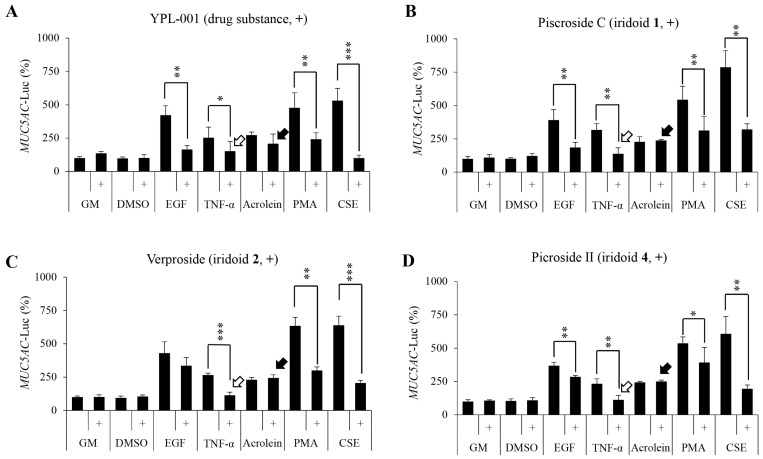
Inhibitory effects of YPL-001 and its three iridoids on various stimuli for mucus overproduction. Inhibitory effects of YPL-001 (**A**), piscroside C (**B**), verproside (**C**), and picroside II (**D**) on various stimuli for MUC5AC production were assayed in HEK293T cells expressing a luciferase reporter gene fused to the MUC5AC promoter (MUC5AC promoter-*Luc*). The cells were pre-treated for 2 h with the indicated concentrations of YPL-001 (20 µM) or the three iridoids (10 µM), followed by treatment with EGF (100 ng/mL), TNF (20 ng/mL), acrolein (30 nM), PMA (100 nM), or CSE (10 µg/mL) for another 16 h. Luciferase activity was significantly reduced by YPL-001, iridoids **1** (piscroside C), **2** (verproside), and **4** (picroside II) in TNF-stimulated cells (white arrows). YPL-001 and its iridoids had no suppressive effect on acrolein-induced inflammation (black arrows). * *p* < 0.05, ** *p* < 0.01, and *** *p* < 0.001, as compared to each stimulus alone. CSE, cigarette smoke extract.

**Figure 4 ijms-24-07229-f004:**
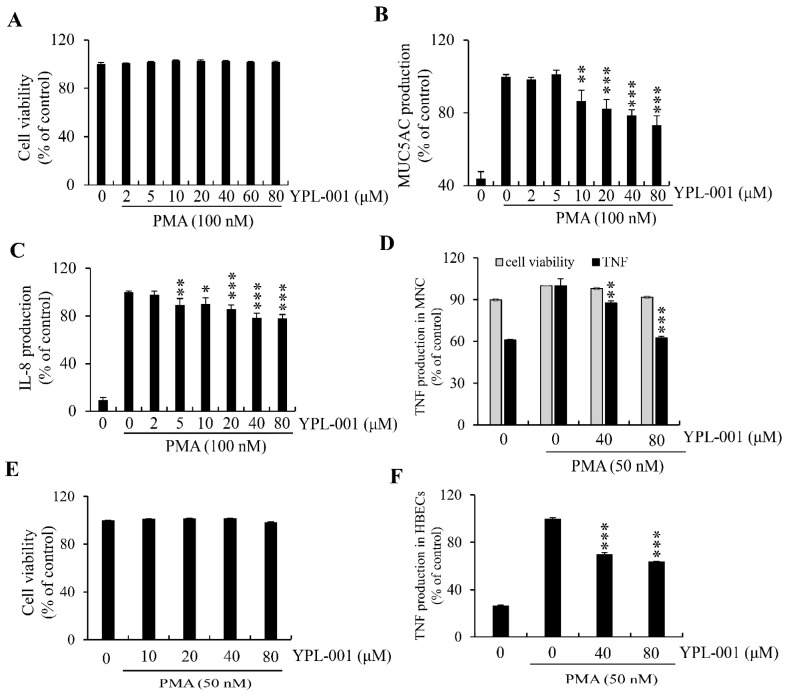
Inhibitory effect of YPL-001 on PMA-induced MUC5AC, IL-8, or TNF secretion in NCI-H292, human mononuclear cells(MNCs), or primary human bronchial epithelial cells (HBECs). (**A**) No cytotoxicity in NCI-H292 treated with YPL-001 at concentrations below 80 μM in the presence of PMA (100 nM). PMA was used to activate PKC. (**B**,**C**) The effect of YPL-001 on PMA-induced secretion of MUC5AC or IL-8 protein was assayed using ELISA. (**D**) Cell viability and TNF secretion by YPL-001 on PMA (50 nM)-stimulated human MNCs isolated from umbilical cord blood (UCB) are measured by CCK-8 and ELISA, respectively. (**E**,**F**) Cell viability and TNF secretion by YPL-001 on PMA (50 nM)-exposed human HBECs are measured by CCK-8 and TNF ELISA, respectively. * *p* < 0.05, ** *p* < 0.01, and *** *p* < 0.001, as compared to the PMA alone.

**Figure 5 ijms-24-07229-f005:**
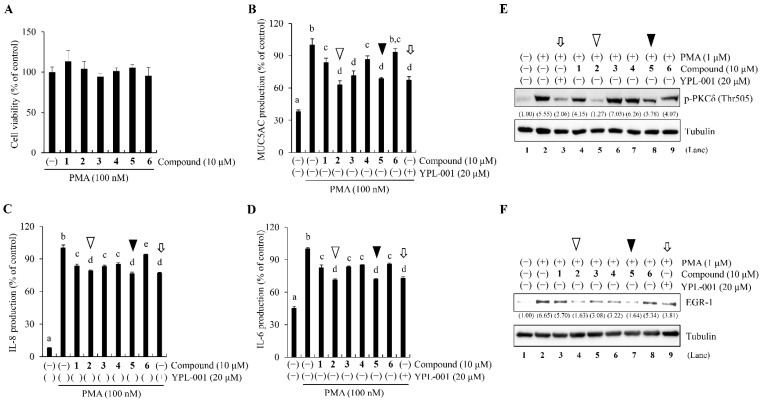
Inhibitory effects of YPL-001 and its six iridoids on PMA/PKCδ/EGR-1-induced inflammation in NCI-H292. (**A**) No cytotoxicity in NCI-H292, treated with the six iridoids in the presence of PMA. (**B**–**D**) YPL-001 or each iridoid suppresses the PMA-induced secretion of MUC5AC, IL-8, and IL-6 proteins. The inhibitory effects of iridoids **2** (white arrowheads) and **5** (black arrowheads) are comparable to that of YPL-001 (white arrow). Two-way ANOVA with Tukey’s multiple comparisons test was performed (**B**–**D**). ^a–e^ Different superscript letters on the bar graph mean significant statistical differences (*p* < 0.05). There is no statistical significance between groups sharing the same letter. (**E**,**F**) The inhibitory effect of iridoids **2** (verproside) and **5** (6-*O*-veratroyl catalpol) on PKCδ phospho-activation (**E**) and its downstream-target EGR-1 expression (**F**). The PKCδ phospho-activation level and EGR-1 expression were assayed using western blotting. Tubulin is a loading control. The relative intensity (a fold of control) from two independent experiments is shown below the band.

**Figure 6 ijms-24-07229-f006:**
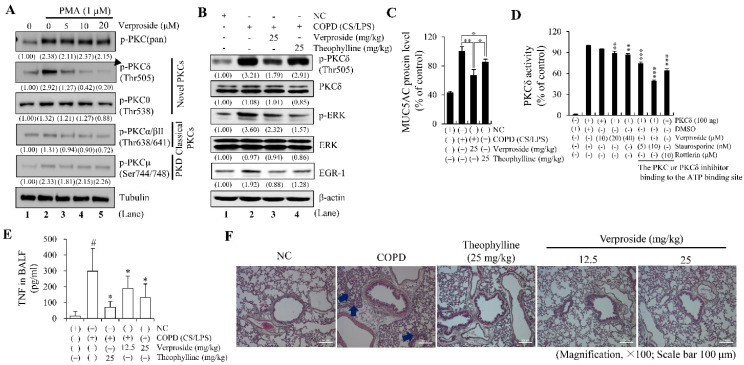
Inhibitory effect of PKCδ activation by verproside, a major component of YPL-001 in vitro and in vivo. (**A**) Verproside suppresses PKCδ activation. In PMA-stimulated NCI-H292 cells, activation of PKC isozymes was analyzed using western blotting of whole-cell lysates using antibodies against phosphorylated forms of PKC isozymes, such as pan PKC, PKCδ, PKCθ, PKCα/βII, and PKCµ. PMA-induced PKCδ activation/phosphorylation is markedly reduced by adding verproside in a concentration-dependent manner (arrowhead). Tubulin is a loading control. (**B**) In the lung tissues of CS/LPS-exposed COPD mice, the phospho-PKCδ, phospho-ERK, and EGR-1 levels increase (lane 2). Verproside effectively reduced activation of the PKCδ/ERK/EGR-1 pathway (lane 3). The suppression of PKCδ by verproside is more effective than that by theophylline. β-actin is a loading control. The relative band intensity (a fold of control) from two independent experiments is shown below the western blot band. (**C**) MUC5AC levels in the lung tissues of COPD model mice were measured using ELISA. (**D**) In vitro PKCδ kinase assay. The inhibitory effects of verproside on PKCδ activity were evaluated using purified PKCδ proteins. Staurosporine and rottlerin are PKC inhibitors that directly bind to the ATP-binding site. ** *p* < 0.01 and *** *p* < 0.001 compared to PKCδ only. (**E**) The levels of TNF in BALF of COPD model mice were measured using ELISA. (**F**) Hematoxylin and eosin (H&E) staining was performed to assess the accumulation of inflammatory cells around the peribronchial region. The inflammatory cells were highly enriched in the peribronchial tissues of COPD mice (blue arrows). NC, normal control mice; COPD, CS, and LPS-exposed mice; Verproside, CS and LPS-exposed mice treated with verproside (12.5 or 25 mg/kg); Theophylline: CS and LPS-exposed mice treated with theophylline (25 mg/kg). ^#^ Significantly different from NC, *p* < 0.01; * *p* < 0.05 and ** *p* < 0.01, compared to the COPD mice group, *n* = 6. Scale bar 100 μm.

**Figure 7 ijms-24-07229-f007:**
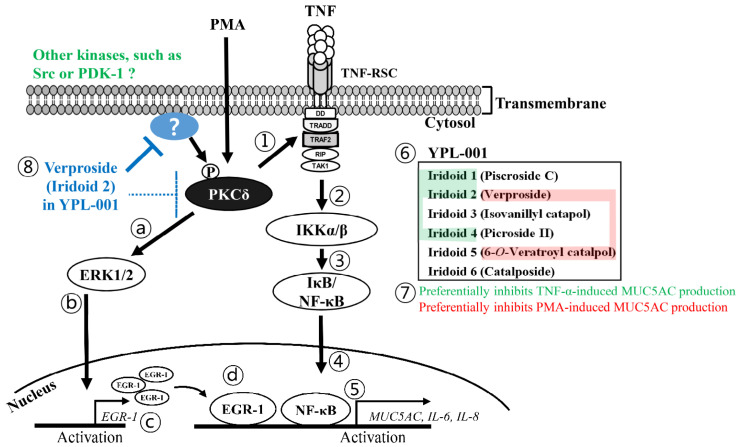
The proposed role of verproside in airway inflammation. In human airway cells, PKCδ elicits the expression of inflammatory genes, such as MUC5AC, IL-6, and IL-8, by regulating two pathways: TNF-induced NF-κB signaling (steps 1–5) and PMA-induced PKCδ/EGR-1 signaling (steps a-d). (Step 1) Activated PKCδ promotes TNF-RSC formation by binding to TRAF2 subunits [5]. (Steps 2–4) Subsequently, TNF-RSC activates IκB kinase (IKK), thereby inducing the nuclear translocation of cytosolic NF-κB by inhibiting IκB activity. (Step 5) Nuclear NF-κB binds to the promoter regions of its target genes to induce MUC5AC overexpression or cytokine release. (Steps a–c) At the same time, activated PKCδ also upregulates EGR-1 expression via ERK1/2. (Step d) EGR-1 cooperates with nuclear NF-κB to upregulate mucin overexpression and cytokine release. (Steps 6–7) YPL-001 efficiently suppressed TNF-induced NF-κB and PMA-induced PKCδ/EGR-1 signaling. Among the six iridoids in YPL-001, iridoids **1**, **2**, and **4** preferentially inhibited the TNF-induced NF-κB pathway (shaded green). In contrast, iridoids **2** and **5** preferentially inhibited PMA/EGR-1-induced inflammation (shaded red). (Step 8) Iridoid **2** efficiently suppresses both NF-κB and EGR-1 activity by inhibiting PKCδ activation. Verproside binds to PKCδ but weakly inhibits the kinase activity (a broken line) compared to its inhibitory effect in NCI-H292 cells or in vivo. Thus, we suppose that verproside may inhibit more upstream components that regulate PKCδ phosphorylation.

## Data Availability

The original contributions presented in the study are included in the article/Appendix A, and further inquiries can be directed to the corresponding authors.

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
