# Peer review of "Verproside, the Most Active Ingredient in YPL-001 Isolated from Pseudolysimachion rotundum var. subintegrum, Decreases Inflammatory Response by Inhibiting PKCδ Activation in Human Lung Epithelial Cells"

_ijms, 2023, doi:10.3390/ijms24087229_

Round 1

Reviewer 1 Report

The paper by Kim et al. describes the anti-inflammatory effects of the iridoid Verproside derived from the medicinal plant Pseudolysimachion rotundum var. subintegrum in the context of COPD. Overall, the paper is well-written, and the experiments are well-designed and rigorous. This is an important area of research in the preclinical molecular sciences, however, there are concerns about the physiological relevance of the data reported in the manuscript.

Minor concerns:

1.    Tumor Necrosis Factor is now only TNF. TNF-a is now an obsolete term.

https://en.wikipedia.org/wiki/Tumor_necrosis_factor

Major concerns:

1.    NCI-H292 cells are pulmonary mucoepidermoid carcinoma cells. Aggressive cancer cell lines inherently exhibit resistance to cytotoxic compounds of all types; compounds which would exert greater cytotoxicity on normal, primary human cells. It is not accurate to refer to 292 cells as ‘human lung epithelial cells’. Therefore, it is difficult to accept the data presented on cytotoxicity using 292 cells as evidence that these compounds will not be cytotoxic in human airways at doses required to exert therapeutic benefit. There is a slight decrease in cell numbers shown in Figure 1A and, for example, iridoid 4 shows a downward trend in cell numbers with increasing dose in Figure 2A. The claims about lack of cytotoxicity would be much stronger if the cytotoxicity assays were performed with proliferating expansion cultures of primary human tracheobronchial (TBEC) and/or small airway epithelial cells (SAEC), and in differentiated air-liquid interface cultures. There are numerous commercial sources for the cells and required media, and there are well-established methods for driving physiological differentiation in air-liquid interface culture.

2.    The authors should repeat a minimal cohort of experiments measuring MUC5AC mRNA and protein secretion using primary human airway cells. It is possible that the effects will be more dramatically beneficial in differentiated primary airway epithelium. 292 cells are notorious for hypersecretion of mucus, and this could explain why even the highest dose of YPL-001 used for the studies shown in Figure 1 only produces an approximately 50% reduction in MUC5AC.

3.    The same comments in major concerns 1 and 2 apply to data shown in Figure 4, especially the IL-8 secretion data. The results presented are informative, but it is possible that the magnitude of both IL-8 induction and inhibition by YPL-001 will be markedly different in normal, primary human airway cells.

4.    The effect of YPL-001 in immune cells is perhaps more important for inflammatory protection in the lung. The authors could strengthen their conclusions about applicability to human COPD by including in vitro PMA stimulation experiments with human PBMC, looking at cytokine/chemokine production as a measure of the anti-inflammatory effects of YPL-001 and individual iridoids.

5.    The mouse model of COPD is essentially an acute lung injury model driven by toxic inhalation exposure over a short period of time. While long-term structural damage associated with COPD/emphysema will not be present after 1 week of exposure, the evidence of the inflammatory injury should be clear histologically with heavy infiltrate. Therapeutic benefit should be visible and perhaps quantifiable on the level of lung histology. There is no inclusion of tissue sections in the main paper or the supplement. At a minimum, it would be greatly beneficial to see H&E sections from each of the experimental groups to gauge protection of the lungs in the treatment groups. IHC staining for important classes of immune cells and tissue homogenate assays for myeloperoxidase would provide deeper insights. The MUC5AC data shown in Figure 6 is informative but does not inform the reader about the effect of Verproside on the inflammatory reaction. Inhibition of PKC-delta activity is expected to reduce the inflammatory response, but this should be shown explicitly in some form.

Author Response

Reviewer #1: The paper by Kim et al. describes the anti-inflammatory effects of the iridoid Verproside derived from the medicinal plant Pseudolysimachion rotundum var. subintegrum in the context of COPD. Overall, the paper is well-written, and the experiments are well-designed and rigorous. This is an important area of research in the preclinical molecular sciences, however, there are concerns about the physiological relevance of the data reported in the manuscript.

Major concerns:

Q1. NCI-H292 cells are pulmonary mucoepidermoid carcinoma cells. Aggressive cancer cell lines inherently exhibit resistance to cytotoxic compounds of all types; compounds which would exert greater cytotoxicity on normal, primary human cells. It is not accurate to refer to 292 cells as ‘human lung epithelial cells’. Therefore, it is difficult to accept the data presented on cytotoxicity using 292 cells as evidence that these compounds will not be cytotoxic in human airways at doses required to exert therapeutic benefit. There is a slight decrease in cell numbers shown in Figure 1A and, for example, iridoid 4 shows a downward trend in cell numbers with increasing dose in Figure 2A. The claims about lack of cytotoxicity would be much stronger if the cytotoxicity assays were performed with proliferating expansion cultures of primary human tracheobronchial (TBEC) and/or small airway epithelial cells (SAEC), and in differentiated air-liquid interface cultures. There are numerous commercial sources for the cells and required media, and there are well-established methods for driving physiological differentiation in air-liquid interface culture.

â–¶ We absolutely agree with the reviewer’s constructive comments. To address the reviewer’s concerns, we currently are performing to add the results of the anti-inflammatory effect by drug treatment using primary human bronchial epithelial cells (HBECs) from normal donors, However, Human primary cells, culture media, and growth reagents arrived at last week (May 3, 2023) in Lonza company. As you know, this experiment requires more time to perform expansion and differentiation using HBECs (For 14-21 days). Thus, we unfortunately were unable to add these results by the revision deadline date (May 15, 2023). However, we will continuously perform experiments according to the reviewer's valuable suggestion.

Q2. The authors should repeat a minimal cohort of experiments measuring MUC5AC mRNA and protein secretion using primary human airway cells. It is possible that the effects will be more dramatically beneficial in differentiated primary airway epithelium. 292 cells are notorious for hypersecretion of mucus, and this could explain why even the highest dose of YPL-001 used for the studies shown in Figure 1 only produces an approximately 50% reduction in MUC5AC.

â–¶ Our response is the same as the reviewer's answer in Q1. If you give us a chance, we will add whether YPL-001 and compounds affects the MUC5AC secretion in TNF- or PMA-stimulated HBECs.

Q3. The same comments in major concerns 1 and 2 apply to data shown in Figure 4, especially the IL-8 secretion data. The results presented are informative, but it is possible that the magnitude of both IL-8 induction and inhibition by YPL-001 will be markedly different in normal, primary human airway cells.

â–¶ Our response is the same as the reviewer's answer in Q1. If you give us a chance, we will add whether YPL-001 affects the IL-8 secretion in PMA-stimulated HBECs.

Q4. The effect of YPL-001 in immune cells is perhaps more important for inflammatory protection in the lung. The authors could strengthen their conclusions about applicability to human COPD by including in vitro PMA stimulation experiments with human PBMC, looking at cytokine/chemokine production as a measure of the anti-inflammatory effects of YPL-001 and individual iridoids.

â–¶ We absolutely agree with the reviewer’s constructive comments. According to the reviewer’s advice, we performed TNF ELISA to check whether YPL-001 affects secretion levels of TNF in PMA-stimulated primary mononuclear cells (MNCs) isolated from human umbilical cord blood. In this result, we showed that YPL-001 suppressed increased TNF levels in PMA-stimulated human MNCs without affecting cell viability. These revised new data were added in Figure 4D described in the revised manuscript (p.6, lines 210-21, 213-214, 218-220; p.12, lines 460-465; p.13, lines 480-482; p.14, lines 550-553 from the top).

Note that mononuclear cells (MNCs) isolated from human umbilical cord blood collected from healthy women with full-term pregnancies, with the agreement of mothers. The study was conducted in accordance with the Declaration of Helsinki, and entire experimental procedures was approved by the Korea Research Institute of Bioscience and Biotechnology (KRIBB) Institutional Review Board (P01-201610-31-002).

Q5. The mouse model of COPD is essentially an acute lung injury model driven by toxic inhalation exposure over a short period of time. While long-term structural damage associated with COPD/emphysema will not be present after 1 week of exposure, the evidence of the inflammatory injury should be clear histologically with heavy infiltrate. Therapeutic benefit should be visible and perhaps quantifiable on the level of lung histology. There is no inclusion of tissue sections in the main paper or the supplement. At a minimum, it would be greatly beneficial to see H&E sections from each of the experimental groups to gauge protection of the lungs in the treatment groups. IHC staining for important classes of immune cells and tissue homogenate assays for myeloperoxidase would provide deeper insights. The MUC5AC data shown in Figure 6 is informative but does not inform the reader about the effect of Verproside on the inflammatory reaction. Inhibition of PKC-delta activity is expected to reduce the inflammatory response, but this should be shown explicitly in some form.

â–¶ Thank you for valuable comments. To address the reviewer's concerns, we newly added results of assessing the inflammatory response using hematoxylin and eosin stain (H&E) tissue sections and bronchoalveolar lavage fluid (BALF) of lung from an experimental mouse model of COPD. In these results, we found that verproside exerts anti-inflammatory action through suppression of inflammatory cell recruitment in lung tissue and TNF secretion in BALF of an experimental mouse model of COPD. These revised new data were added in Figure 6E and Figure 6F described in the revised manuscript (p.8, lines 286-290; p.9, lines 320-324; p.13, lines 507-512; p.14, lines 540-548 from the top).

Minor concerns:

Q1.  Tumor Necrosis Factor is now only TNF. TNF-a is now an obsolete term.

https://en.wikipedia.org/wiki/Tumor_necrosis_factor

â–¶ As noted by the reviewer, we have corrected TNF-a to TNF in full text. The revised part of the manuscript including changed Figure 7 can be checked through the “change tracking” of the document.

Reviewer 2 Report

To the authors

This is an interesting study analysing the therapeutic value of YPL-001, a natural drug that has already undergone clinical trials, in inflammatory lung disease. The authors investigate the role of the different active ingredients in this drug and propose the mechanism of action of the specific iridoid that displays the most potent anti-inflammatory effect.

The study is well designed, the experiments clearly explained, performed and analysed and the manuscript clearly written. It is a comprehensive body of work that explores the anti-inflammatory effects of this natural drug and extensively investigates the potential mechanism of action using both in vitro cell models and in vivo mouse models of lung disease and a range of technologies. Although more investigations are required, the current study convincingly demonstrates that use of this natural drug (or its most active ingredient) could be more beneficial to patients with inflammatory lung disease such as COPD than standard medications.  

Specific comments:

1.      Overall, this is elegant work, however, the in vitro experiments were performed using an epithelial cell line derived from Mucoepidermoid pulmonary (bronchial) carcinoma. Would the authors comment on the choice of this cell line.

2.       Although cell lines are important tools allowing extensive manipulations when performing mechanistic studies, primary airway cells from COPD patients would provide more valuable supportive evidence of the anti-inflammatory effect /use of this drug treatment.

3.       Would the authors comment on the protocol used to produce the mouse COPD model, the physiological relevance of 8cigs/day and the relatively small group numbers.

4.       The labelling in Figure 2 is a little confusing. The small letters on top of bars are not explained adequately in the legend. Also, panels E-G would benefit from being labelled with the iridoid number, not just the name, for consistency with the rest of panels in the figure.

Author Response

Reviewer #2: This is an interesting study analysing the therapeutic value of YPL-001, a natural drug that has already undergone clinical trials, in inflammatory lung disease. The authors investigate the role of the different active ingredients in this drug and propose the mechanism of action of the specific iridoid that displays the most potent anti-inflammatory effect. The study is well designed, the experiments clearly explained, performed and analysed and the manuscript clearly written. It is a comprehensive body of work that explores the anti-inflammatory effects of this natural drug and extensively investigates the potential mechanism of action using both in vitro cell models and in vivo mouse models of lung disease and a range of technologies. Although more investigations are required, the current study convincingly demonstrates that use of this natural drug (or its most active ingredient) could be more beneficial to patients with inflammatory lung disease such as COPD than standard medications. 

Specific comments:

Q1. Overall, this is elegant work, however, the in vitro experiments were performed   using an epithelial cell line derived from Mucoepidermoid pulmonary (bronchial) carcinoma. Would the authors comment on the choice of this cell line.

â–¶ We have two reasons to use the NCI-H292 cell line for our in vitro experiments.

1) NCI-H292 is an in vitro model cell line widely used to study human lungs and various pulmonary diseases such as airway inflammation (Lee et al., 2018), COPD (Yang et al., 2021), and asthma (Lee et al., 2021). Thus, we choose the NCI-H292 line for our in vitro screen.

2) Mucus overproduction/hypersecretion is one of the pathological characteristics of COPD. Among human cell lines in ATCC, NCI-H292 is the only cell line registered as a human pulmonary mucoepidermoid cell. As mucoepidermoid cells, the NCI-H292 cells readily produce and secret mucus proteins, including MUC5AC in vitro conditions. Moreover, various stimuli can boost this mucus production/secretion in NCI-H292 cells (Birru et al., 2021;  Song et al., 2017), mimicking inflammatory conditions in human airway cells (Cao et al., 2021).        

Since the primary goal of our experiment was to find a natural compound reducing mucus overproduction/hypersecretion in COPD conditions, we take advantage of this feature of NCI-H292 cells for in vitro COPD drug screening.

We have concisely added the above information (#1 and #2) on the “Results” section in our revised manuscript (on page 2, lines 94-96). We sincerely hope this response satisfies the reviewer’s request.

References

Birru, R.L., Bein, K., Wells, H., Bondarchuk, N., Barchowsky, A., Di, Y.P., Leikauf, G.D., 2021. Phloretin, an Apple Polyphenol, Inhibits Pathogen-Induced Mucin Overproduction. Mol Nutr Food Res 65, e2000658.

Cao, X., Coyle, J.P., Xiong, R., Wang, Y., Heflich, R.H., Ren, B., Gwinn, W.M., Hayden, P., Rojanasakul, L., 2021. Invited review: human air-liquid-interface organotypic airway tissue models derived from primary tracheobronchial epithelial cells-overview and perspectives. In Vitro Cell Dev Biol Anim 57, 104-132.

Lee, B.W., Ha, J.H., Ji, Y., Jeong, S.H., Kim, J.H., Lee, J., Park, J.Y., Kwon, H.J., Jung, K., Kim, J.C., Ryu, Y.B., Lee, I.C., 2021. Alnus hirsuta (Spach) Rupr. Attenuates Airway Inflammation and Mucus Overproduction in a Murine Model of Ovalbumin-Challenged Asthma. Front Pharmacol 12, 614442.

Lee, S.U., Ryu, H.W., Lee, S., Shin, I.S., Choi, J.H., Lee, J.W., Lee, J., Kim, M.O., Lee, H.J., Ahn, K.S., Hong, S.T., Oh, S.R., 2018. Lignans Isolated From Flower Buds of Magnolia fargesii Attenuate Airway Inflammation Induced by Cigarette Smoke in vitro and in vivo. Front Pharmacol 9, 970.

Song, W.Y., Song, Y.S., Ryu, H.W., Oh, S.R., Hong, J., Yoon, D.Y., 2017. Tilianin Inhibits MUC5AC Expression Mediated Via Down-Regulation of EGFR-MEK-ERK-Sp1 Signaling Pathway in NCI-H292 Human Airway Cells. J Microbiol Biotechnol 27, 49-56.

Yang, N., Singhera, G.K., Yan, Y.X., Pieper, M.P., Leung, J.M., Sin, D.D., Dorscheid, D.R., 2021. Olodaterol exerts anti-inflammatory effects on COPD airway epithelial cells. Respir Res 22, 65.

Q2. Although cell lines are important tools allowing extensive manipulations when performing mechanistic studies, primary airway cells from COPD patients would provide more valuable supportive evidence of the anti-inflammatory effect/use of this drug treatment.

â–¶ We absolutely agree with the reviewer’s constructive comments. Based on the reviewer's valuable comments, we performed TNF ELISA to check whether YPL-001 affects secretion levels of TNF in PMA-stimulated primary mononuclear cells (MNCs) isolated from human umbilical cord blood, although not primary airway cells isolated from COPD patients. In this result, we showed that YPL-001 suppresses increased TNF levels in PMA-stimulated human MNCs without affecting cell viability. These revised new data were added in Figure 4D described in the revised manuscript (p.6, lines 210-211, 213-214, 218-220; p.12, lines 460-465; p.13, lines 480-482; p.14, lines 550-553 from the top).

Currently, we are performing to add the results of the anti-inflammatory effect by drug treatment using primary human bronchial epithelial cells (HBECs) from normal donors, although not primary airway cells isolated from COPD patients. However, Human primary cells, culture media, and growth reagents arrived at last week (May 3, 2023) in Lonza company. Moreover, we must perform differentiation experiment test (For 14-21 days) under ALI condition using Human primary cell. Thus, we currently couldn't add these results.

Q3. Would the authors comment on the protocol used to produce the mouse COPD model, the physiological relevance of 8cigs/day and the relatively small group numbers.

â–¶ Thank you for valuable comments. To describe the physiological relevance of 8 cigs/day in an experimental mouse model of COPD, we have newly performed hematoxylin and eosin stain (H&E) stain in lung tissue, and cytokine analysis in BALF of lung. At this time, the administration of verporside were 12.5 and 25 mg/kg. In these results, we showed that the whole-body exposure to 8 cigs/day for seven days using a smoking machine can cause lung inflammation. Furthermore, we showed that administration of verproside exerts anti-inflammatory action through suppression of TNF secretion in BALF and inflammatory cell recruitment in lung tissue of an experimental mouse model of COPD. These revised new data were added in Figure 6E and Figure 6F described in the revised manuscript (p.9, lines 320-324; p.13, lines 507-512; p.14, lines 540-548 from the top).

Q4. The labelling in Figure 2 is a little confusing. The small letters on top of bars are not explained adequately in the legend. Also, panels E-G would benefit from being labelled with the iridoid number, not just the name, for consistency with the rest of panels in the figure.

â–¶ According to the reviewer’s suggestion, we have described the small letters in legends of Figure 2 and Figure 5. Different superscript letters on the bar graph mean significant statistical differences (p<0.05) as a result of Tukey's multiple comparison. There is no statistical significance between groups sharing the same letter. We revised legends of Figure 2 and Figure 5 in the revised manuscript (p.4, lines 137-138 ; p.7, lines 251-252). 

Round 2

Reviewer 1 Report

I greatly appreciate the authors' earnest efforts to address my comments and improve the quality of their study. 

Author Response

Thank you for your thoughtful comments. Based on the reviewer's valuable suggestions, we will continue to conduct experiments using differentiation of human primary cells under ALI. In the next opportunity, we will definitely apply this to improve the quality of our research.

Reviewer 2 Report

Thank you to the authors for addressing most comments in a satisfactory manner. The manuscript is much improved, with minor edits required in the figures (some writing looks distorted and/or semi-covered). However, i still believe that additional data using primary human airway cells would be of great value to this work. As the authors indicate having already obtained the primary cells, i would recommend performing experiments similar to the MNC cells (IL-8 production, TNF secretion) to add to this manuscript.   

Author Response

Reviewer #2: Thank you to the authors for addressing most comments in a satisfactory manner. The manuscript is much improved, with minor edits required in the figures (some writing looks distorted and/or semi-covered). However, I still believe that additional data using primary human airway cells would be of great value to this work. As the authors indicate having already obtained the primary cells, i would recommend performing experiments similar to the MNC cells (IL-8 production, TNF secretion) to add to this manuscript.   

â–¶ We absolutely agree with the reviewer’s constructive comments. To address the reviewer’s concerns, we newly performed ELISA to check whether YPL-001 affects secretion levels of TNF in PMA-stimulated primary human bronchial epithelial cells (HBECs) from normal donors. In this result, we showed that YPL-001 significantly suppresses increased TNF levels in PMA-stimulated human HBECs without affecting cell viability. These revised new data were added in Figure 4E and Figure 4F and described in the revised manuscript (p.6, lines 213-214, 217-218, p.7, lines 222-224; p.12, lines 472-474 and 478; p.13, lines 486-489 from the top).

The revised part of the manuscript including changed Figure 4 can be checked through the “change tracking” of the document.
